# Determinants of Obtaining COVID-19 Vaccination among Health Care Workers with Access to Free COVID-19 Vaccination: A Cross-Sectional Study

**DOI:** 10.3390/vaccines10010039

**Published:** 2021-12-29

**Authors:** Mariam R. Elkhayat, Maiada K. Hashem, Ahmed T. Helal, Omar M. Shaaban, Ahmed K. Ibrahim, Taghreed S. Meshref, Hussein Elkhayat, Mohamed Moustafa, Mohammed Nahed Attia Mohammed, Azza M. Ezzeldin, Hebatallah G. Rashed, Alaa Bazeed, Islam H. Ibrahim, Ahmed Mokhtar Mahmoud, Moaiad Eldin Ahmed Mohamed, Reem Sayad, Shimaa A. Elghazally

**Affiliations:** 1Occupational and Environmental Medicine Department, Assiut University, Assiut 71526, Egypt; alaabazeed@med.aun.edu.eg (A.B.); shima_dola@aun.edu.eg (S.A.E.); 2Chest Disease Department, Faculty of Medicine, Assiut University, Assiut 71526, Egypt; maiada.hashem@aun.edu.eg; 3Social Casework Department, Faculty of Social Work, Assiut University, Assiut 71514, Egypt; ahmed.socialwork@aun.edu.eg; 4Sociology and Social Work Department, College of Arts and Social Science, Sultan Qaboos University, Muscat 123, Oman; 5Department of Obstetrics and Gynecology, Faculty of Medicine, Assiut University Hospital, Assiut 71526, Egypt; omshaaban2000@aun.edu.eg; 6Public Health Department, Faculty of Medicine, Assiut University, Assiut 71526, Egypt; ahmed.ibrahim@aun.edu.eg; 7Critical Care Unit, Department of Internal Medicine, Faculty of Medicine, Assiut University, Assiut 71526, Egypt; tagreed.meshref@aun.edu.eg; 8Cardiothoracic Surgery Department, Faculty of Medicine, Assiut University, Assiut 71526, Egypt; elkhayat@aun.edu.eg; 9Orthopedics and Trauma Surgery, Faculty of Medicine, Assiut University, Assiut 71526, Egypt; mohamed.m.alaa@aun.edu.eg; 10Oral & Maxillofacial Surgery, Faculty of Dentistry, Assiut University, Assiut 71526, Egypt; mohammed_nahed@aun.edu.eg; 11Clinical Pathology Department, Assiut University Hospital, Assiut 71526, Egypt; azzam80@aun.edu.eg (A.M.E.); h.rashed64@aun.edu.eg (H.G.R.); 12Faculty of Medicine, Assiut University, Assiut 71526, Egypt; Islam.16266300@med.aun.edu.eg (I.H.I.); ahmedmokhtarmahmoud98@gmail.com (A.M.M.); moide1998@gmail.com (M.E.A.M.); reemoo.8527@gmail.com (R.S.)

**Keywords:** health care workers, COVID-19, vaccine

## Abstract

Introduction: Despite global efforts to contain the illness, COVID-19 continues to have severe health, life, and economic repercussions; thus, maintaining vaccine development is mandatory. Different directions concerning COVID-19 vaccines have emerged as a result of the vaccine’s unpredictability. Aims: To study the determinants of the attitudes of healthcare workers (HCWs) to receiving or refusing to receive the vaccine. Methods: The current study adopted an interviewed questionnaire between June and August 2021. A total of 341 HCWs currently working at Assiut University hospitals offered to receive the vaccine were included. Results: Only half of the HCWs (42%) accepted the COVID-19 vaccine. The most common reason that motivated the HCWs was being more susceptible than others to infection (71.8%). On other hand, the common reasons for refusing included: previously contracted the virus (64.8%); did not have time (58.8%); warned by a doctor not to take it (53.8%). Nearly one-third of nonaccepting HCWs depended on television, the Internet, and friends who refused the vaccine for information (*p* < 0.05). In the final multivariate regression model, there were six significant predictors: sex, job category, chronic disease, being vaccinated for influenza, and using Assiut University hospital staff and the Ministry of Health as sources of information (*p* < 0.05). Conclusion: Misinformation and negative conceptions are still barriers against achieving the desired rate of vaccination, especially for vulnerable groups such as HCWs.

## 1. Introduction

Egypt ranks fourth in terms of population, holding 105 million people based on projections of the latest United Nations data, with an Asyut population of 420,585. In 2018, the median age of the population reached 23.9 years. Based on 2017 estimates, there are about 37.6% between 25–54 years of age (the working age) [1], with the presence of about 100,000 doctors and 132,000 nurses working in governmental hospitals all over Egypt [2]. However, the CFR percentage in Egypt (5.92%), and, as Dr. Hala Zayed, the Minister of Health and Population, confessed, the COVID-19 infection rate among the Egyptians is much more than what the government has quantified. This is attributed to the high numbers of infected cases in Egypt which pass as officially undetected and recovered at home [3].

Despite global efforts to contain the illness (physical separation, face masking, travel restrictions, and quarantine), COVID-19 continues to have severe health, life, and economic repercussions [4]. The world’s aspirations are tied to an effective preventive strategy and vaccination, which has demonstrated its potential to prevent illnesses and save lives over time [5]. Several vaccines began to emerge near the end of 2020. There are approximately 100 potential vaccines, with the most widely distributed six candidate vaccines presently in the third phase of testing. Their composition, storage needs, and effectiveness are all different (ranging from 70.4% to 95%) [6]. Different directions, views, and attitudes concerning COVID-19 vaccines have emerged as a result of the vaccine’s unpredictability. Governments and public health specialists face issues because of these discrepancies. Vaccination apprehension has been named one of the top ten global health challenges by the World Health Organization (WHO) [7]. Many studies from the United States of America (USA) [8], China [9], the United Kingdom, Ireland [10], and Congo [11] found varying levels of vaccination acceptability and hesitation among the general public and healthcare staff. The greatest rates of reluctance were among participants from the western areas of the Arab world according to a multicentric research study including Arab nations that measured HCWs’ trepidation about vaccination (Egypt, Tunisia, Algeria, and Morocco) [12].

Since December 2020, Egypt began to receive shipments of anti-COVID-19 vaccines, such as Sinopharm (BBIBP-CorV), AstraZeneca vaccine, and Sputnik V. Priority groups for vaccination are (A) medical staff at quarantine, fever, and chest hospitals; (B) patients with cancer, kidney, or immunity problems, chronic disease patients, and the elderly; (C) and eventually all citizens above 18 years of age. As of March 2021, Egypt has started COVID-19 vaccine rollouts. The country aims to vaccinate 40% of its population against COVID-19 by the end of 2021 [1].

We must quantify vaccine reluctance and understand the specific grounds behind it in order to tackle the projected impending issue of vaccination hesitancy.

Recently, and as a part of the national program of vaccination, the governmental health authority offered an extreme priority and free COVID-19 vaccination to all those who are accepting from the Assiut University Hospital, including doctors, nurses, workers, and employees.

This work is designed to study the attitudes and the determinants of the attitudes of those groups of the population receiving or refusing to receive their first dose of vaccination.

## 2. Materials and Methods

### 2.1. Study Design and Site

A cross-sectional study was conducted among healthcare workers currently working in Assiut University hospitals (Dental, Obstetrics and Gynecology, main building, and Cardiac) and who were offered to receive the first dose of AstraZeneca and the Sinopharm vaccine. This study aimed to determine the attitudes and to assess the main contributing factors for receiving or refusing to receive the first dose of vaccination. The study was conducted in the period between June and August 2021.

Sample size calculation was carried out using the EPI Info 2000 statistical package. The calculation was based on an expected frequency for acceptance among Egyptian healthcare workers to the COVD-19 vaccine (21%) [3] during a COVID-19 attack, with a difference of 5% and a confidence interval of 95%. The minimum required sample was 255. The sample was raised to 341 HCWs to compensate for refusals and incomplete data. The sampling technique was multistage, where in the first stage there was a simple random sampling so to choose 4 hospitals from 8 hospitals, i.e., a sample in each hospital was proportionate to the size of the number of HCWs working in it (the total in 4 hospitals was 7233, where the main hospital represented 46% of the total, and we took 159 HCWs, which represented the same percentage from the total sample). Then, in each hospital a convenience sampling technique was applied, as all HCWs had same call and chance for obtaining their vaccine (Figure 1). A questionnaire was applied via the interview method.

### 2.2. Data Collection Tools

A predesigned interviewed questionnaire was prepared for the assessment of the attitudes and the determinants of the attitudes of these groups of the population for receiving or refusing to receive their first dose of vaccination.

The questionnaire was composed of different parts, as follows:

Personal history, including name, age, residency, marital status, and comorbidity conditions such as diabetes mellitus (DM) and hypertension (HTN).

Previous history of infection with COVID-19 within 3 months or the infection of a family member, and the exposure assessment was detected by asking about the number of attendances in COVID-19 isolation areas and the number of exposures to confirmed COVID-19 patients.

Asking the HCWs who were offered the COVID-19 vaccination if they accepted it or not, and the determinants of their decision. HCWs who accepted were asked 17 questions collected from previous studies [3,12], and each had to answer “yes”, “no”, or “maybe”, with a Cronbach’s alpha of 0.709. Nonaccepting HCWs were asked questions collected from previous studies [13,14], and each had to answer “yes”, “no”, or “maybe”, with a Cronbach’s alpha of 0.829.

The degree of acceptance of the HCWs who accepted vaccination was determined by two main questions which were asked to the participants: (i) their willingness to be vaccinated themselves and (ii) their willingness to recommend the vaccines to their patients, using a five-point scale from ‘no, certainly not’ and ‘no, probably not’ to ‘yes, probably’, and ‘yes, certainly’, with a ‘do not know’ option. A score of the presumptive acceptance of future COVID-19 vaccines was constructed based on the responses to the two questions in the survey about COVID-19 vaccines. The score was derived from awarding points per participant depending on the different possible responses given, with zero points for an answer ‘no, certainly not’, one point for ‘no, probably not’, two points for ‘yes, probably’, and three points for ‘yes, certainly’. ‘Do not know’ answers did not get any points and were considered separately. The points obtained for each of the two questions per participant were then summed to obtain the score (Cronbach’s alpha: 0.83; range: 0–6). We then used the score to categorize participants according to their degree of ‘COVID-19 vaccine acceptance’: ‘high acceptance’ (score > 4), ‘moderate acceptance’ (score = 4), and ‘hesitancy or reluctance (to score < 4, or answers ‘do not know’ to at least one of the two questions) [15].

The opinion of the HCWs toward the system of vaccine application, which was illustrated to all HCWs offered the vaccine, was rated on the Likert scale from 0 to 10 and was classified as bad (0 to 4), moderate (5 and 6), and good (7 to 10).

### 2.3. Procedures

A thorough literature review was conducted by the authors for the questionnaire design. For data collection, 4 training settings were performed by qualified researchers, then the pilot study was done with only 10 questionnaires and was not included in the analysis so to test the questionnaire questions and the time consumed. Moreover, during data collection, several sudden audit visits were performed by trainee researchers to data collectors so to assure the data collection. Then, the questionnaire was filled out by the trained data collector from the assigned HCWs at Assiut University hospitals by interviewing them. Each interview took about 15–20 min, after discussing the aim of research and confirming their acceptance to share in the research.

### 2.4. Statistical Tests

Data was analyzed using the SPSS software package version 21 (IBM-SPSS Inc., Chicago, IL, USA). Descriptive statistics were performed as frequency and cross tabulations for categorical variables. The Chi-square test was used to compare the independent categorical variables. Logistic regression was carried out to evaluate the possible attitude correlates. The significant *p*-value was set at a 0.05 cutoff.

## 3. Results

### 3.1. Sociodemographics, Job Categories, Chronic Diseases, Previous COVID-19 Infection and Exposure, and Main Sources of Information Differences among Accepting and Nonaccepting HCWs to COVID-19 Vaccine

The current research adopted an observational cross-sectional design to determine the attitude of HCWs regarding the COVID-19 vaccine. A total 341 completed questionnaires were included for the final analysis. The mean participants’ age was about 34 ± 9.27. The female/male ratio was about 1:2 (38%/62%). Likely, the single to married ratio was 1:2. Most of them (59%) had an urban residence. The cohort of HCWs were classified as 37% nurses, 31% doctors, 13.8% administrators, 12% workers and assistant nurses, and 6% pharmacists and technicians (Table 1).

All sociodemographic characteristics (age, sex, residence, and marital status) were a significant predictor for accepting the vaccine. Young (18–<30), male, urban, and single HCWs had a higher acceptance. Moreover, job category had significant effect on vaccine acceptance, as two-thirds of the accepting HCWs were physicians (60.6%) versus only 16.2% nurses and 4.9% workers and assistant nurses. A previous history of receiving an influenza vaccine was an important predictor for COVID-19 vaccination, as two-thirds (66.3%) of those nonaccepting of the vaccine were vaccinated from influenzas, versus half of those accepting the COVID-19 vaccine were not vaccinated from influenzas (*p* < 0.05). Regarding exposure to COVID-19, having shifts in COVID-19 isolation areas had no significant effect on accepting the COVID-19 vaccine (*p* > 0.05), while the number of exposures of HCWs to confirmed COVID-19 patients had a significant effect on their acceptance to the COVID-19 vaccine, as 55.6% of accepting HCWs were exposed more than three times to COVID-19 patients vs. 32.2% of nonaccepting HWCs who were exposed more than three times. nearly one-third of the nonaccepting HCWs depend on television, the Internet, and friends who refuse vaccine for information (*p* < 0.05); on the opposite side, nearly 60% the accepting HCWs depend on international organizations and the Assiut medical staff for information (*p* < 0.05) (Table 1).

### 3.2. Frequency of Accepting and Nonaccepting Cohort of HCWs toward COVID-19 Vaccine, and Frequency of Regret of HCWs among the Accepting Ones

Figure 2 revealed that nearly half of the HCWs, 42% (95% CI: 38: 46), accepted the COVID-19 vaccine, and only 5% (95% CI: 3:7) of them felt regret.

### 3.3. Predictors of Vaccine Acceptance among Accepted HCWs

The most common reasons that motivated the HCWs to obtain their COVID-19 vaccine were: being more susceptible than others to infection (71.8%); vaccination is a collective action to prevent the spread of disease (68.3%); the vaccine will help me not to get the virus (64.8%); the vaccine will help me not to infect those around me with the virus (64.8%); the benefits of the vaccine are much greater than its harm (60.6%), while the lowest cause was receiving the vaccine from the recommendation of a family member (16.8%) (Table 2). The majority (83.5%) had a high degree of acceptance toward the COVID-19 vaccine (95% CI: 81:86). Moreover, two-thirds (65.7%) stated that the system of vaccine application was good.

### 3.4. Predictors of Vaccine Nonacceptance among Nonaccepted HCWs

On the other hand, Table 3 revealed what the most common reasons for refusing the COVID-19 vaccine were: recently contracted the virus and do not need the vaccine (64.8%); I do not have enough time to take the vaccine (58.8%); warned by a doctor not to take the vaccine (53.8%); the infection with the coronavirus is not so severe that I should receive the vaccine (49.7%); however, the lowest cause concerned the side effects/complications that may happen to me if I take the vaccine (3%) (Table 3).

### 3.5. Multivariable Regression Analysis for the Main Predictors for Accepting COVID-19 Vaccine among the Offered Cohort of HCWs

Table 4 displayed the predictors of acceptance among the studied cohort. A total of 20 important predicting factors (with 16 significant factors by univariate analysis; Chi-square) and four other nonsignificant factors were added in the model, as having shifts in the COVID-19 isolation area, the presence of chronic diseases, using the Ministry of Health website, and reading lectures, due to their importance. After 11 models by backward, the final model contained eight predictors, where the significant predictors were: sex, job category, history of chronic disease, obtaining influenza vaccines, and the Assiut University hospital staff and Ministry of Health as the source of information. The HCWs vaccinated with influenza vaccines were 2.34 times more liable to accept the COVID-19 vaccine in comparison with those HCWs not previously vaccinated (AOR = 2.34, 95% CI: 1.02–5.38, *p*-value < 0.05), and this was statistically significant. Regarding the history of chronic disease, HCWs with chronic diseases were two times more liable to accept the COVID-19 vaccines in comparison with the healthy HCWs (AOR = 2.34, 95% CI: 1.02–5.38, *p*-value < 0.05), and this was statistically significant. Likewise, HCWs who depend on Assiut University staff as trusted sources of information were two times more liable to accept the COVID-19 vaccines (AOR = 2.29, 95% CI: 1.25–4.21, *p*-value < 0.05), and this was statistically significant. On the other hand, females were 62% more liable to refuse the vaccine than males (AOR = 0.38, 95% CI: 0.20–0.73, *p*-value < 0.05).

## 4. Discussion

Healthcare workers and their households are at a higher risk of getting COVID-19 infection [16,17]. With the absence of curative treatment for COVID-19 infection, vaccinating HCWs against COVID-19 is critical. However, variable degrees of acceptance to vaccination were reported [18]. In this study, nearly half (42%) of the recruited HCWs accepted the vaccination. The majority (83.5%) of them had a high acceptance degree toward the COVID-19 vaccine.

Earlier studies from Egypt reported lower acceptance rates, ranging from 21% [4] to 26% [19]. Interestingly, the acceptance rate was reported to be 34.9% among Egyptian medical students [12]. The higher rate in the current study could be explained by the time of the data collection. Previous studies were carried out prior to start of the COVID-19 vaccination program in Egypt. Later, with actual vaccine administration, the availability of vaccines, easy access to the service, satisfaction about the vaccination application process, and tranquility regarding the short-term safety might contribute to increasing the acceptance rate.

This rate is also higher than others reported earlier in many low-, mid-, and even high-income countries in different geographic areas [11,13,14,20]. On the other hand, some studies have reported higher acceptance rates up to 70–90% [21,22,23,24,25]. In line with our study, Qattan et al. [26] reported a moderate acceptance rate towards COVID-19 vaccination in KSA.

In agreement with most of the published literature [11,24,27,28], physicians showed more acceptance to vaccination than other HCWs in the current study (60.6% acceptance among physicians compared to 16.2% and 4.9% among nurses and workers, respectively). Generally speaking, higher levels of education are associated with a more positive attitude towards vaccination [29,30,31].

Age was not a predictor for COVID-19 vaccination acceptance in this study. However, younger participants (18–30 years old) had significantly higher acceptance rates. A study from Italy [32] reported associations between younger age and the intention to get vaccinated. Conversely, older age was found to have a significant association with a positive attitude towards COVID-19 vaccination in most of the published literature [13,14,19,27,33], which may be explained by the more severe impact of the pandemic with increasing age, and that older unvaccinated people are more likely to be hospitalized or die from COVID-19 [18]. This disagreement might be explained by the relative younger mean age of HCWs included in our studies (34 ± 9.27 years), with only 8.5% of participants 50 years of age or more. Thus, the impact of older age with more comorbid diseases was not well estimated. However, we found that HCWs with chronic diseases were two times more liable to accept the COVID-19 vaccines in comparison with healthy workers.

Female healthcare workers were found to be 62% more liable to refuse the vaccine than the male workers included in the current study, which agreed with most previous studies [13,14,34]. This was likely due to their higher fear of side effects such as infertility, serious side effects making them unable to take care of their families, or greater susceptibility to myths and misinformation from media. On the other hand, males had better health-seeking behaviors and appreciation of advice about COVID-19 vaccines [18,35].

In line with a former report [21], healthcare workers who were previously vaccinated with the influenza vaccine were 2.34 times more liable to accept the COVID-19 vaccine in the current study. The behavior, knowledge, and attitude of HCWs toward influenza vaccination seemed to be similar to that toward COVID-19, being accepting or opposing to vaccination in general [35,36]. Moreover, some HCWs might believe in the role of the influenza vaccine in decreasing the risk of severe COVID-19 infection, besides reducing the risk of a coinfection of influenza and COVID-19 [37].

Investigators and analysts had focused on knowledge as a crucial factor controlling the behavior and attitudes of HCWs towards vaccination [38,39,40]. The source of the data influenced the degree to which the recipients incorporated this data into their decision-making process [41,42,43]. Healthcare workers’ explanations or recommendations have a positive influence on the attitude toward the vaccine [21]. In the present study, HCWs who depend on Assiut University staff as a trusted source of information were two times more liable to accept the COVID-19 vaccines, with nearly 60% of accepting HCWs depending on international organizations and Assiut medical staff. Trust in the institutions from which information about vaccines was obtained is an essential driver of vaccine acceptance, for the general population and HCWs as well [43].

On the other hand, nearly two-thirds of HCWs depend on television, the Internet, and friends who refused the vaccine. These results agreed with previous a previous study from Egypt [19]. Moreover, unlike most of the previous publications which reported that concerns about the safety, efficacy, and side effects of the COVID-19 vaccines were the most prevalent reasons for vaccination refusal and hesitancy [4,26,44,45], those were the lowest frequent causes to refuse vaccination in our study. Misinformation and false perceptions, including “I recently contracted the virus and do not need the vaccine; I do not have enough time to take the vaccine; I was warned by a doctor not to take the vaccine; the infection with the coronavirus is not so severe that I should receive the vaccine”, were the most frequent causes. Thus, the impact of social media cannot be neglected, especially in less developed countries, nor can taking in the danger of the spread of misinformation across medias, which the WHO has named the “infodemic” [46].

In view of the foregoing, the negative attitude supported by misinformation possess an extra burden on the Egyptian government to achieve mass vaccination. Tailored programs encompassing different scientific and social modalities must be implemented in order to reach this target.

Even though the results of the current study express the actual situation regarding the attitude toward COVID-19 vaccination in Egypt, as all the accepting participants already received at least one dose of their scheduled vaccination and the study was performed by an interviewed, not online, questionnaire, it still has some limitations. The cross-sectional design of the study does not allow to follow the changing behavior of the participants toward the vaccine and its predictors. Besides, the study represents the attitude in a single healthcare facility in Upper Egypt. Thus, further prospective multicentric studies are recommended.

## 5. Conclusions

Healthcare worker attitudes toward COVID-19 vaccination in Egypt is improving, with higher acceptance rates than the prevaccination period. However, misinformation and negative conceptions are still barriers against achieving the desired rate of vaccination. Education and tailored interventions should be implemented to ensure that healthcare workers are vaccinated with the available COVID-19 vaccines.

## 6. Strengths and Limitations

The COVID-19 vaccination program is one of the most important concerns in the world for caring for and facing COVID-19 infection. Hesitency toward the COVID-19 vaccine is an important burden, so it is important to study, especially among the most vulnerable categories such as HCWs. One of the important limitations of this study is the duration of the exposure per visit of any COVID-19 case, which is an important factor that was not assessed, and a sampling technique which influences the representativeness of the results.

## Figures and Tables

**Figure 1 vaccines-10-00039-f001:**
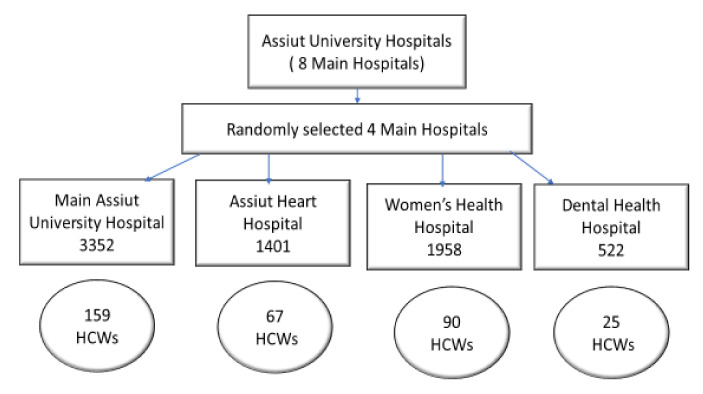
Flow chart of recruited healthcare workers in Assiut University hospitals.

**Figure 2 vaccines-10-00039-f002:**
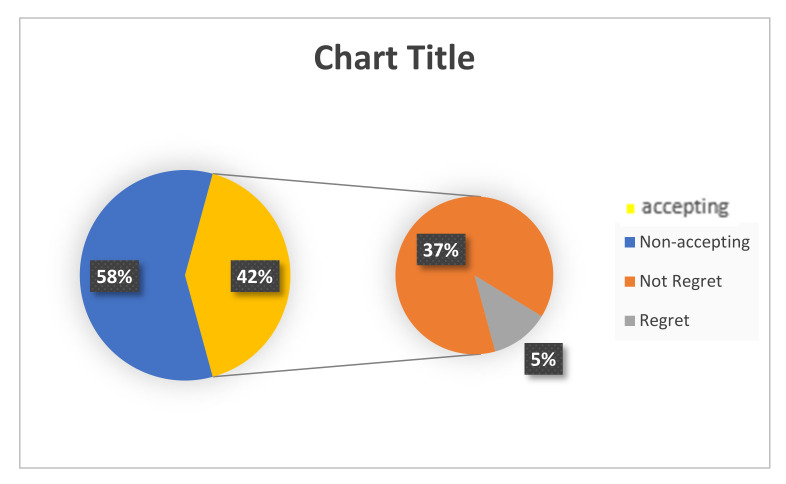
Frequency of accepting and nonaccepting cohort HCWs toward COVID-19 vaccine, and frequency of regret among the accepting HCWs.

**Table 1 vaccines-10-00039-t001:** Sociodemographic, job categories, chronic diseases, previous COVID-19 infection and exposure, and main sources of information differences among accepting and nonaccepting HCWs to COVID-19 vaccine (N = 341).

Characteristics	N (%)	Accepting N = 142	Nonaccepting N = 199	*p*-Value *
Age categories				
18–<30	145 (42.5%)	74 (52.1%)	71 (35.7%)	**0.003**
30–<50	167 (49.0%)	54 (38.0%)	113 (56.8%)	
50–61	29 (8.5%)	14 (9.9%)	15 (7.5%)	
Sex				
Male	131 (38.4%)	89 (62.7%)	42 (21.1%)	**<0.001**
Female	210 (61.6%)	53 (37.3%)	157 (78.9%)	
Residence				
Rural	86 (25.2%)	24 (16.9%)	62 (31.2%)	**0.006**
Urban	202 (59.2%)	97 (68.3%)	105 (52.8%)	
Rural/Urban	53 (15.5%)	21 (14.8%)	32 (16.1%)	
Marital status				
Single	126 (37.0%)	73 (51.4%)	53 (26.6%)	**<0.001**
Married	215 (63.0%)	69 (48.6%)	146 (73.4%)	
Job categories				
Doctors	106 (31.1%)	86 (60.6%)	20 (10.1%)	**<0.001**
Nurses	126 (37.0%)	23 (16.2%)	103 (51.8%)	
Workers/Assistant nurses	41 (12%)	7 (4.9%)	34 (17.1%)	
Admin	47 (13.8%)	19 (13.4%)	28 (14.1%)	
Pharmacists, lab technicians, and chemists	21 (6.2%)	7 (4.9%)	14 (7.0%)	
Hospitals				
Main Assiut University Hospital	159 (46.6%)	69 (48.6%)	90 (45.2%)	0.104
Assiut Heart Hospital	67 (19.6%)	32 (22.5%)	35 (17.6%)	
Women Health Hospital	90 (26.4%)	36 (25.4%)	54 (27.1%)	
Dental Health Hospital	25 (7.3%)	5 (3.5%)	20 (10.1%)	
History of chronic disease	50 (14.7%)	20 (14.1%)	30 (15.1%)	0.799
History of previous COVID-19 infection	77 (22.6%)	36 (25.4%)	41 (20.6%)	0.301
History of family COVID-19 infection	77 (22.6%)	43 (30.3%)	34 (17.1%)	0.004
History of influenza vaccination				
Yes	133 (39%)	66 (46.5%)	67 (33.7%)	**0.017**
No	208 (61%)	76 (53.5%)	132 (66.3%)	
Had shifts in COVID-19 isolation area				
Never	221 (64.8%)	93 (65.5%)	128 (64.3%)	0.123
Once	33 (9.7%)	8 (5.6%)	25 (12.6%)	
Twice	11 (3.2%)	3 (2.1%)	8 (4.0%)	
Three times	2 (0.6%)	1 (0.7%)	1 (0.5%)	
More than three times	74 (21.7%)	37 (26.1%)	37 (18.6%)	
Times exposed to confirmed COVID-19 patients				
Never	116 (34.0%)	34 (23.9%)	82 (41.2%)	**<0.001**
Once	51 (15.0%)	15 (10.6%)	36 (18.1%)	
Twice	24 (7.0%)	11 (7.7%)	13 (6.5%)	
Three times	7 (2.1%)	3 (2.1%)	4 (2.0%)	
More than three times	143 (41.9%)	79 (55.6%)	64 (32.2%)	
Attend COVID-19 vaccine awareness sessions in Assiut University hospitals				
Yes	64 (18.8%)	30 (21.1%)	34 (17.1%)	**<0.001**
No	167 (49.0%)	52 (36.6%)	115 (57.8%)	
Never hear about it	110 (32.3%)	60 (42.3%)	50 (25.1%)	
Main source of scientific information regarding COVID-19 vaccine				
International organization	140 (41.1%)	81 (57.0%)	59 (29.6%)	**<0.001**
Assiut medical staff	173 (50.7%)	82 (57.7%)	91 (45.7%)	**0.029**
Television	90 (26.4%)	19 (13.4%)	71 (35.7%)	**<0.001**
Friends	73 (21.4%)	19 (13.4%)	54 (27.1%)	**0.002**
Research papers	73 (21.4%)	43 (30.3%)	30 (15.1%)	**0.001**
Facebook and social media	102 (29.9%)	31 (21.8%)	71 (35.7%)	**0.006**
Health Ministry website	65 (19.1%)	21 (14.8%)	44 (22.1%)	0.159
Scientific lectures	38 (11.1%)	16 (11.3%)	22 (11.1%)	0.951

* Chi-square test, bold *p*-values were significant (*p* < 0.05).

**Table 2 vaccines-10-00039-t002:** Predictors of vaccine acceptance among accepted HCWs (N = 142).

Predictors for Vaccine Acceptance (N = 142)	Yes	No	“Maybe”
Is taking vaccine is mandatory and practical	83 (58.5%)	27 (19.0%)	32 (22.5%)
I think the vaccine will help me not get the virus	92 (64.8%)	8 (5.6%)	42 (29.6%)
I think that the vaccine will help me not to infect those around me with the virus	92 (64.8%)	13 (9.2%)	37 (26.1%)
I suffer from a chronic disease, so I took the vaccine for fear of my life	29 (20.4%)	92 (64.8%)	21 (14.8%)
My trust in the manufacturer of the vaccine that I took encouraged me to take the vaccine	24 (16.9%)	42 (29.6%)	76 (53.5%)
I took the vaccine because I was afraid that I would not get it in the future	38 (26.8%)	51 (35.9%)	53 (37.3%)
I took the vaccine because it was recommended by international scientific agencies	74 (52.1%)	19 (13.4%)	49 (34.5%)
I believe that there are no serious side effects from taking the vaccine	37 (26.1%)	38 (26.8%)	67 (47.2%)
I think that the benefits of the vaccine are much more than its harm	86 (60.6%)	11 (7.7%)	45 (31.7%)
I took the vaccine as a recommendation of a doctor I trust	51 (35.9%)	40 (28.2%)	51 (35.9%)
I took the vaccine as a recommendation of a trusted friend/person	35 (24.6%)	49 (34.5%)	58 (40.8%)
I took the vaccine as a recommendation of a member of my family	23 (16.2%)	68 (47.9%)	51 (35.9%)
I believe that vaccination is a collective action to prevent the spread of disease	97 (68.3%)	9 (6.3%)	36 (25.4%)
I am confident that the public authorities decide the best interest for society	63 (44.4%)	18 (12.7%)	61 (43.0%)
The continuous increase of cases around me made me not hesitate to take the vaccine	84 (59.2%)	11 (7.7%)	47(33.1%)
I feel that I am more susceptible than others to infection due to the nature of my work	102 (71.8%)	11 (7.7%)	29 (20.4%)
What I heard and saw on the Internet and social networks about the vaccine encourage me take it	57 (40.1%)	31 (21.8%)	54 (38.0%)

**Table 3 vaccines-10-00039-t003:** Predictors of vaccine nonacceptance among nonaccepted HCWs (N = 199).

Predictors for Vaccine Nonacceptance (N = 199)	No	Yes	“Maybe”
I think, if I took the vaccine, I might catch COVID-19 infection	40 (20.1%)	89 (44.7%)	70 (35.2%)
I don’t think there is an effective vaccine in avoiding infection with the virus COVID-19	29 (14.6%)	100 (50.3%)	70 (35.2%)
I am concerned about the side effects/complications that may happen to me if I take the vaccine	6 (3.0%)	147 (73.9%)	46 (23.1%)
I refused because of the request to sign for the responsibility in case I was given a vaccine	83 (41.7%)	69 (34.7%)	47 (23.6%)
My lack of confidence in the healthcare system followed in my country	71 (35.7%)	64 (32.2%)	64 (32.2%)
I do not trust the manufacturers of this vaccine	45 (22.6%)	88 (44.2%)	66 (33.2%)
I think that the emerging coronavirus is a threat that has been amplified, and therefore there is no need to take a vaccine	93 (46.7%)	39 (19.6%)	67 (33.7%)
I am a precautionary person, so I do not need to take the vaccine	42 (21.1%)	79 (39.7%)	78 (39.2%)
I recently contracted the virus and do not need the vaccine	129 (64.8%)	32 (16.1%)	38 (19.1%)
I was warned by a doctor not to take the vaccine	107 (53.8%)	49 (24.6%)	43 (21.6%)
I received a warning from someone around me not to take the vaccine	79 (39.7%)	74 (37.2%)	46 (23.1%)
What I heard and saw on the Internet and social networks about the vaccine made me reject it	58 (29.1%)	86 (43.2%)	55 (27.6%)
The difficulty of the procedures followed in taking the vaccine prevented me from taking it	81 (40.7%)	45 (22.6%)	73 (36.7%)
My lack of confidence in the safety of vaccines, as there are no proven studies so far on the benefits and side effects of the vaccine	25 (12.6%)	107 (53.8%)	67 (33.7%)
My immune system is very strong, it also protects me from diseases, and I do not need a vaccine	79 (39.7%)	40 (20.1%)	80 (40.2%)
I do not have enough time to take the vaccine	117 (58.8%)	29 (14.6%)	53 (26.6%)
I think the infection with the coronavirus is not so severe that I should receive the vaccine	99 (49.7%)	38 (19.1%)	62 (31.2%)
My fear of getting an infection while taking the vaccine prevented me from taking it	50 (25.1%)	85 (42.7%)	64 (32.2%)
The safety of a vaccine developed in an emergency, during an epidemic, cannot be guaranteed	33 (16.6%)	102 (51.3%)	64 (32.2%)
My belief in the superiority of acquiring immunity against infectious diseases naturally, instead of taking the vaccine	47 (23.6%)	83 (41.7%)	69 (34.7%)

**Table 4 vaccines-10-00039-t004:** Multivariable regression analysis for the main predictors for accepting the COVID-19 vaccine among offered cohort HCWs.

Variable	OR (95% CI)	*p*-Value	AOR (95% CI)	*p*-Value
Age categories	1.02 (0.98:1.06)	0.463		
Sex (F vs. M)	0.48 (0.24:0.96)	0.038	0.38 (0.20:0.73)	**0.003**
Residence (R vs. U and mixed)	1.29 (0.79:2.11)	0.308		
Marital status (S vs. M)	0.49 (0.24:1.01)	0.052	0.54 (0.27:1.04)	0.065
Job categories (Doctors)	Ref [1]	**<0.001**	Ref [1]	**<0.001**
Nurses	0.05 (0.2:0.15)	<0.001	0.08 (0.03:0.18)	**<0.001**
Workers/assistant nurses	0.04 (0.01:0.14)	<0.001	0.05 (0.02:0.14)	**<0.001**
Admin	0.23 (0.07:0.71)	0.010	0.27 (0.11:0.69)	**0.007**
Pharmacists, lab technicians, and chemists	0.09 (0.03:0.33)	<0.001	0.14 (0.04:0.42)	**0.001**
History of chronic disease	2.42 (0.99:5.88)	0.052	2.34 (1.02:5.38)	**0.046**
History of previous COVID-19 infection	0.59 (0.27:1.29)	0.187		
History of family COVID-19 infection	0.81 (0.36:1.78)	0.596		
History of influenzas’ vaccination	3.12 (1.64:5.93)	0.001	2.83 (1.56:5.15)	**0.001**
Had shifts in COVID-19 isolation area	0.65 (0.32:1.31)	0.232		
Exposed to confirmed COVID-19 patients	2.13 (1.05:4.32)	0.038		
Attending COVID-19 vaccine awareness sessions in Assiut University hospitals	1.27 (0.55:2.94)	0.574		
Main source of information regarding COVID 19 vaccine				
International organization	1.21 (0.57:2.54)	0.618		
Assiut medical staff	2.30 (1.19:4.45)	0.013	2.29 (1.25:4.21)	**0.008**
Television	0.79 (0.35:1.73)	0.556		
Friends	0.58 (0.26:1.34)	0.205	0.53 (0.25:1.19)	0.096
Research papers	0.62 (0.27:1.44)	0.270		
Facebook and social media	0.79 (0.37:1.71)	0.561		
Health Ministry website	0.47 (0.21:1.05)	0.066	0.48 (0.23:1.01)	**0.054**
Scientific lectures	2.19 (0.81:5.91)	0.124		

F: female; M: male; R: rural; U: urban; S: single; M: married. bold *p*-values were significant (*p* < 0.05).

## Data Availability

Data is available upon request for ethical purposes.

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
