# Peer review of "Determinants of Obtaining COVID-19 Vaccination among Health Care Workers with Access to Free COVID-19 Vaccination: A Cross-Sectional Study"

_vaccines, 2021, doi:10.3390/vaccines10010039_

Round 1

Reviewer 1 Report

The manuscript is well written but i find it sometimes a bit overemphatic: for example the first sentence "COVID-19 is a surprise attack in which the entire planet joins forces and fights with knowledge and discipline." i am not sure it adds anything but one could argue that parts of the planets have not used knowledge and discipline...

i think it would be useful to have a summary of the actual burden of covid in egypt what it meant in terms of health services saturation (i don't know was it as bad as in europe or the usa?). also the demogrphic structure of egypt (median age) proportion of persons being overweight. not in detail but i presume these factors are part of the context in which health professionals weigh their calculations....

furthermore it would be useful to have information about national campaigns did the government, the health authorities, medical nursing societies actively communicate on vaccine, these contextual elements are essential for those who do not know egypt. are there any mandates ?

Figure 2  is not clear

the yellow color is what? the legend should be more self explanatory

fig 3 takes a lot of space and could be scrapped for just text. same for fig 4 which is not self explanatory

table 1 remove bullet points

table 2 & 3 "maybe"

please specify the logistic regression modelling strategy: how were the variables selected for the multivariate analysis was there a strategy to aim for the most parsimonious model of not. goodness of fit test?

Reviewer 2 Report

Extensive English language edit is essential. Multiple typing errors shall be corrected.

In sampling technique, the sample size per each hospital is not proportionate to size, questioning the stratification.

Sample selection in each hospital was convenient however according to the different attitudes and type of workers, that was essential to be stratified random selection. 

The difference between 0 point for an answer ‘no, certainly not’,  not get any point for ‘Do not know’ answers is not clear although authors reported the Regret group.

Duration of exposure per visit of any COVID19 case, is another important factor, not assessed and shall be discussed as a limitation in discussion.

Using a trained data collector/s, make measurement less valid and reliable in comparison with self-administered questionnaires, the reason and the limitation shall be discussed and the way used for quality control and assurance in the study specifically for this issue and the number of trained data collectors, shall be discussed.

In table 1, all percentages for vaccine accepted and not-accepted groups should be based on columns like the total.

Using two different questionnaires for Predictors of Vaccine Acceptance and vaccine not-acceptance, make it impossible to perform any comparative analysis and inference. This is a major limitation.

Comparison between hospitals shall be performed, not only for the outcomes but also for the determinants and predictors.

95% confidence intervals shall be reported for main estimates of vaccine acceptance and high acceptance.

Decision: Major revision.

Round 2

Reviewer 1 Report

The authors have answered my queries.

Reviewer 2 Report

Thanks for changes performed and explanations.

Limitations in sampling technique which influence the representativeness of the results should be discussed in the limitations.

Comparison of different hospitals should be performed to assess any significant difference between HCWs attitude in each hospital.

In table 1, in the 1st column N, percentages provided correctly, for example percentage of each age category in total sample. But in assessing the accepting and non-accepting groups, in next two columns, the frequency of accepting and non-accepting provided in each variable state, for example percentages of accepting and non-accepting provided in 18-30 years old category. The correct form is providing frequency of variable states in accepting and non-accepting like first N column. For example frequency of each age category in accepting, and in non-accepting groups.

Decision, Major revision.

Author Response

Point 1: Limitations in sampling technique which influence the representativeness of the results should be discussed in the limitations.

Response 1:  the following was added in limitation in line 335

Point 2: Comparison of different hospitals should be performed to assess any significant difference between HCWs attitude in each hospital.

Response 2:  the following was added in Table 1, there was no statistically significant difference (p = 0.104)

Point 3: In table 1, in the 1st column N, percentages provided correctly, for example percentage of each age category in total sample. But in assessing the accepting and non-accepting groups, in next two columns, the frequency of accepting and non-accepting provided in each variable state, for example percentages of accepting and non-accepting provided in 18-30 years old category. The correct form is providing frequency of variable states in accepting and non-accepting like first N column. For example, frequency of each age category in accepting, and in non-accepting groups.

Response 3: thanks for your appreciated comment, for descriptive purposes, we calculated column percentage for the total sample as there is no row percentage and on comparing accepting versus non accepting, we calculated row percentages as we aim to compare the prevalence of acceptance in each age group to each other .

Round 3

Reviewer 2 Report

For 
Point 3: In table 1, in the 1st column N, percentages provided correctly, for example percentage of each age category in total sample. But in assessing the accepting and non-accepting groups, in next two columns, the frequency of accepting and non-accepting provided in each variable state, for example percentages of accepting and non-accepting provided in 18-30 years old category. The correct form is providing frequency of variable states in accepting and non-accepting like first N column. For example, frequency of each age category in accepting, and in non-accepting groups.
and the
Response 3: thanks for your appreciated comment, for descriptive purposes, we calculated column percentage for the total sample as there is no row percentage and on comparing accepting versus non accepting, we calculated row percentages as we aim to compare the prevalence of acceptance in each age group to each other .
if so, the not accepting column provides no more information because the number in this column is simple subtraction of accepting from total and the percentage will be the 100 minus the percentage in vaccinated.
If authors insist this type of presentation, it should be with only accepting column with 95% confidence intervals for each estimate, showing significant differences between each sub-group, by omitting the non-accepting. In this from acceptation rates in each category of independent variables will be reported correctly. 

Decision: Major revision.
